# Relationship between Regular Green Tea Intake and Osteoporosis in Korean Postmenopausal Women: A Nationwide Study

**DOI:** 10.3390/nu14010087

**Published:** 2021-12-26

**Authors:** Dan Bi Lee, Hong Ji Song, Yu-Jin Paek, Kyung Hee Park, Young-Gyun Seo, Hye-Mi Noh

**Affiliations:** Department of Family Medicine, Hallym University Sacred Heart Hospital, College of Medicine, Hallym University, Anyang 14068, Korea; tenuto_87@naver.com (D.B.L.); hongji@hallym.or.kr (H.J.S.); paek@hallym.or.kr (Y.-J.P.); beloved@hallym.or.kr (K.H.P.); yg035@hallym.or.kr (Y.-G.S.)

**Keywords:** bone density, green tea, osteoporosis, tea

## Abstract

Mixed results have been reported regarding whether habitual tea intake affects bone health. This study investigated the relationship between green tea intake and bone mineral density (BMD) in postmenopausal Korean women. We used data from the Korean National Health and Nutrition Examination Surveys from 2008 to 2011 and divided the participants into three groups according to their frequency of green tea intake over the past 12 months. BMD of the lumbar spine, total femur, and femur neck was measured using dual-energy X-ray absorptiometry. The odds ratios (ORs) and 95% confidence intervals (CIs) of osteoporosis and osteopenia according to green tea consumption were analyzed. Participants who did not consume green tea or consumed less than one cup per day were more likely to have osteopenia of the lumbar spine or femur than those who consumed it once to three times a day (OR 1.81 and 1.85, 95% CI, 1.20–2.71; and 1.23–2.77). Moreover, ORs for osteoporosis were 1.91 (95% CI 1.13–3.23) and 1.82 (95% CI 1.09–3.05) in non-consumers and consumers who drank less than one cup per day, respectively, compared with the reference group. These results support that green tea consumption may have benefits on bone health.

## 1. Introduction

Osteoporosis is a musculoskeletal disorder characterized by the destroyed structure of bone tissue and reduced bone density [1]. This diminishes bone strength and increases its risk of fracture. Fractures significantly undermine the quality of life and increase medical and social burdens [2,3].

Women have a high risk of osteoporosis after menopause, due to reduced estrogen production, which results in faster bone replacement rates and higher bone absorption. The rate of bone loss is highest among women after menopause with 1.8–2.3% in the spine and 1.0–1.4% in the hip reported per year [4]. Bone mineral density (BMD) is used for the diagnosis of osteopenia or osteoporosis, and this is associated with several genetic and environmental factors. Therefore, it is important to optimize correctable factors to reduce bone loss and osteoporosis risk in such women. Furthermore, diet is an environmental factor that can be modified in these women [5].

Tea is the most consumed drink worldwide [6]. In particular, green tea is the favorite drink in Asia, and about 40–60% of Japanese and Korean adults drink green tea [7]. Green tea contains flavonoids, which possess estrogen-like effects that increase the differentiation of osteoblasts and decrease osteoclastic activity [8]. In particular, previous studies have reported that polyphenols, a representative component of green tea, may increase BMD and reduce bone loss through antioxidant activity [9]. Since the 1990s, many studies have demonstrated the relationship between drinking tea and osteoporosis or BMD. However, there have been mixed results reported. In postmenopausal women, there is a positive relationship between tea intake and BMD, with a lower risk of osteoporosis [10,11,12,13]. It has also been reported that subjects who drink tea every day have a lower risk of hip fracture [14]. Meanwhile, there is no significant association observed between tea consumption and BMD as mentioned in some studies [15,16].

In Korea, there has been no research conducted on the relationship between tea intake and BMD. BMD has high heritability, and this varies with racial and ethnic differences [17]. Regarding the considerable consumption of green tea in Korea, there is a need to elucidate the effect of green tea on BMD in the Korean population. Therefore, we investigated a nationwide cross-sectional study to investigate the relationship between green tea intake and BMD in postmenopausal Korean women. In addition, we explored whether habitual intake of green tea affected the risk of osteoporosis.

## 2. Materials and Methods

### 2.1. Study Population

We used data from the Korean National Health and Nutrition Examination (KNHANES), a survey that was supervised by the Korean Centers for Disease Control and Prevention (KCDC) during 2008–2011. It was a collection of representative samples of non-institutionalized Koreans, and a multistage probability and stratified sampling method were used therein. The survey consisted of health interviews, health examinations, and nutrition surveys. A total of 37,753 participants participated in the study. Among the participants, 20,558 were women, and there were 6438 postmenopausal women over 50 years of age. To exclude early menopause, only women over 50 years of age were selected in the study. Study participants consisted only of women who responded that their menstruation had stopped for more than 12 months, except for those with surgically or artificially induced menopause. After excluding participants without data on green tea consumption and those who did not have dual-energy X-ray absorptiometry (DXA) examination, finally, 3530 postmenopausal women were included in the study.

All study participants provided informed consent before enrolling in the study. KNHANES was reviewed and approved by the KCDC Institutional Review Board (IRB: 2008-04EXP-01-C, 2009-01CON-03-2C, 2010-02CON-21-C, 201102CON-06-C). All procedures performed in the studies involving human participants were performed in accordance with the Declaration of Helsinki.

### 2.2. Assessment of Lifestyle and Anthropometric Parameters

A questionnaire was used to examine smoking status, alcohol intake, physical activity, level of education, household income, medical history, frequency of coffee intake, age at menopause, and use of hormonal replacement therapy in the participants. Smoking status was classified into two groups: current smoker and ex-smoker. Binge drinking was defined as drinking more than five glasses more than once a week. (1) A minimum of 20 min of strenuous physical activity at least 3 times a week, (2) a minimum of 30 min of moderate activity at least 5 times a week, or (3) walking for 30 min at least 5 times a week were defined as regular exercise. The questionnaire also examined whether the participant had comorbidities such as diabetes mellitus, chronic kidney disease, rheumatoid arthritis, thyroid disease, cancer, and liver cirrhosis that could cause secondary osteoporosis. Body mass index (BMI) was calculated based on height and weight measured by an experienced inspector (BMI = weight [kg]/height (m)^2^). Dietary calcium intake, total energy intake, and protein intake were calculated based on the 24-h diet recall method using CAN-Pro software (version 3.0; Korean Nutrition Society, Seoul, Korea).

### 2.3. Measurement of BMD and Definition of Osteopenia and Osteoporosis

Using DXA (DISCOVERY-W fan-beam densitometer; Hologic, Bedford, MA, USA), the BMD of the lumbar spine (L1-L4), femur neck, and total femur were measured. The coefficients of variation of BMD for the lumbar spine, femur neck, and total femur were 1.9%, 2.5%, and 1.8%, respectively. The lumbar spine and left femur were also measured for each participant. The right femur was measured only if it was impossible to measure the left femur due to lesions such as surgery or fracture. If there was only one lumbar vertebra to be analyzed due to surgery, a lumbar examination was not performed for the participant. If the participant had undergone surgery on both the femurs, the femurs were not measured accordingly. A T-score was defined as the standard deviations compared to the reference mean BMD for sex- and race-matched young adults; −2.5 < T-score < −1.0 and T-score ≤ −2.5 were defined as osteopenia and osteoporosis [1].

### 2.4. Assessment of Green Tea Consumption

We used a questionnaire that surveyed the frequency of green tea intake in the past 12 months. It consisted of nine categories: almost no drinking, 6–11 cups (1 cup = 200 mL) a year, 1 cup per month, 2–3 cups a month, 1 cup per week, 2–3 cups a week, 4–6 cups a week, 1 cup a day, 2 cups a day, and 3 cups a day. We then reclassified participants into three groups: non-consumers who responded to ‘almost no drinking’ (*n* = 1893), participants who drank green tea less than once a day (*n* = 1336), and participants who drank one, two, or three cups of green tea per day (*n* = 301).

### 2.5. Statistical Analysis

Complex sample analysis was used to determine the sampling method and response rate. Data are presented as the mean ± standard error or weighted number (weighted %) for continuous or categorical variables. General characteristics of study subjects were compared using one-way analysis of variance for continuous variables and chi-square test for categorical variables. The Kolmogorov-Smirnov test was used for normality of continuous variables. Serum levels of 25-hydroxy vitamin D, dietary intake of calcium, protein, total energy, and frequency of coffee intake did not fit a normal distribution, and we used log-transformed values for comparison. Crude and adjusted least squares mean of BMD were estimated for each green tea group, and a multiple linear regression model was used for a trend test. Multivariate logistic regression analysis for the odds ratios (ORs) and 95% confidence intervals (CIs) were used to evaluate the association between green tea consumption and osteoporosis or osteopenia. First, we calculated crude ORs and adjusted for age (model 1). Model 2 was adjusted for age, BMI, and other characteristics, such as socioeconomic status factors (household income and education level), behavioral factors (smoking and alcohol use), nutritional factors (coffee, calcium, total energy, and protein intake), and hormonal factors (age of menopause and hormone replacement therapy) were adjusted in Model 2. Comorbidities were adjusted in Model 3. Statistical significance level for analysis was set at *p* < 0.05, and all analyses were performed using SPSS (version 20.0; SPSS Inc., Chicago, IL, USA).

## 3. Results

### 3.1. General Characteristics According to Green Tea Intake

Table 1 represents the general characteristics of the study subjects according to green tea intake. Participants who reported drinking 1–3 cups of green tea a day were younger than those who did not (60.93 ± 0.602 vs. 65.47 ± 0.314). The rate of binge drinking among people who drank 1–3 cups of green tea a day was higher than that in other groups (8.3%, 5.1%, and 2.5%, *p* < 0.001), and the proportion of people who exercised regularly was also highest (59.1%, 52.5%, and 48.1%, respectively, *p* = 0.004). Higher education and income levels were associated with a higher consumption of green tea. Dietary calcium intake was lowest in non-green tea consumers and it increased as the amount of green tea intake increased in the participant. The coffee intake was highest in people who drank 1–3 cups of green tea a day. However, the total energy and protein intake was the highest in participants who drank 1< cups of green tea per day (all *p* < 0.001).

### 3.2. Prevalence of Osteoporosis and Osteopenia According to Green Tea Intake

A comparison with the prevalence of osteoporosis and osteopenia among three groups of green tea intake is shown (Table 2). The group that consumed more green tea had more women with normal BMD at all measurement sites. In contrast, if participants drank 1–3 cups of green tea a day, the prevalence of osteopenia and osteoporosis was the lowest at all BMD measurement sites (all *p* < 0.001).

### 3.3. The Crude and Adjusted Means of BMD by Green Tea Intake

The crude and adjusted mean BMD among three groups of green tea intake is shown in Table 3. The participants who had higher consumption of green tea tended to have higher crude means of BMD at all measurement sites (all *p* for trend < 0.001). The age-adjusted means of the lumbar spine, femur neck, and total femur were significantly higher in tea consumers than in non-consumers (*p* for trend 0.001, 0.002, and 0.004). However, fully covariate-adjusted means were not different between tea consumers and non-consumers at either BMD measurement site.

### 3.4. Green Tea Intake and Osteopenia

Table 4 shows the results of multivariable logistic regression analyses to estimate the relationship between green tea intake and osteopenia. Non-consumers and participants who consumed 1< cups of green tea per day had a higher OR for osteopenia, especially of the lumbar spine (OR 1.54, 95% CI 1.10–2.16 and OR 1.63, 95% CI 1.14–2.33) and femur neck (OR 1.50, 95% CI 1.06–2.13 and OR 1.43, 95% CI 1.01–2.04) compared to those who consumed 1–3 cups of green tea a day (reference group). There was no significant association in total femur.

ORs of osteopenia of the lumbar spine, femur neck, or total femur were 1.81 (95% CI 1.20–2.71) and 1.85 (95% CI 1.23–2.77) in the non-consumers and in consumers who drank 1< cups of green tea per day compared to the reference group.

### 3.5. Green Tea Intake and Osteoporosis

Table 5 also shows the relationship between green tea intake and osteoporosis. Neither of the measurement site (lumbar spine, femur neck, or total femur) was significantly different for the risk of osteoporosis among groups of green tea intake. However, compared to the reference group, non-consumers and consumers with less than one cup per day had a higher OR for osteoporosis in the lumbar spine, femur neck, or total femur (OR 1.91, 95% CI 1.13–3.23 and OR 1.82, 95% CI 1.09–3.05).

## 4. Discussion

In this nationwide study, we demonstrated that green tea intake has a positive association with bone health among postmenopausal Korean women. Covariate-adjusted mean BMD did not differ according to green tea intake; however, green tea consumers had a lower likelihood of osteopenia and osteoporosis. This inverse association between green tea intake and osteopenia was more prominent in the lumbar spine and femur neck.

Although each site (lumbar spine, femur neck, or total femur) was not significantly different for the risk of osteoporosis, in clinical practice, osteoporosis can be diagnosed even if only one of the three measurement sites meets the criteria. When we analyzed the lumbar spine, femur neck, or total femur according to the diagnostic criteria, there was a significant inverse association observed between green tea intake and the risk of osteoporosis.

To date, there have been inconsistent results regarding the impact of tea intake on bone health. Some studies have demonstrated a positive association between tea consumption and BMD, and a lower risk of osteoporosis [10,11,12,13], whereas other studies have reported no association between tea intake and BMD [15,16]. Three meta-analyses of observational studies concluded that tea consumption might increase BMD and decrease the risk of osteoporosis [18,19]. There are possible mechanisms by which green tea affects bone metabolism. Green tea contains polyphenols, known as antioxidants, and it has a protective effect on BMD. Animal studies have reported that green tea polyphenols have a high antioxidative activity due to their activation of liver glutathione peroxidase and lower levels of 8-Hydroxy-2-deoxyguanosine [20,21]. The antioxidant capacity of green tea polyphenols can protect osteoblasts from oxidative stress. This leads to increased osteoblast activity and reduced osteoclast formation. These have a conservative effect on bone, accordingly. Green tea polyphenols also contain epigallocatechin gallate (EGCG), which is a key component of tea. It increases alkaline phosphatase activity and promotes bone mineralization [22]. Isoflavonoids are known to have many biological effects, such as weak estrogenic effects. Hernandez et al. and Hoover et al. found that this effect of isoflavonoids in tea can affect BMD in postmenopausal women with low levels of endogenous estrogen; however, it does not affect BMD in premenopausal women or men [23,24]. In laboratory research, catechin has also been suggested to have weak estrogenic effects via estrogen receptors, and it strongly stimulates osteoblast-like cells [25].

In this study, participants who were younger and had higher education and income levels consumed more green tea. This is presumably because relatively younger and employed people tend to drink tea during social relationships, and they purchase green tea-related products as a gift. Moreover, people who have knowledge of the various health benefits of green tea consume more green tea than those who do not. According to a consumer study, health benefits are the main reason that consumers buy green tea [26]. However, after adjusting for age and socioeconomic status, the positive association between green tea intake and bone health remains statistically significant.

In our study, we also demonstrated that women who consumed green tea regularly drank more coffee. In addition to green tea, coffee contains polyphenols such as caffeic acid and chlorogenic acid, which are known to inhibit osteoclastogenesis and bone resorption [27]. However, it can be assumed that the overall caffeine intake of people who consume green tea regularly is higher than that of others. Nevertheless, the prevalence of osteoporosis was the lowest in this group. Caffeine is a key component of both green tea and coffee. Although tea leaves have higher caffeine (2–3 percent) than coffee (1 percent), tea is usually diluted with water to drink. Therefore, the amount of caffeine in tea is less than half of that of coffee. In such a case, the effect of caffeine on BMD may be lower than that of coffee. Caffeine has adverse effects on bone health, because it reduces calcium absorption and promotes the excretion of calcium into urine and feces [28]. However, since the absorption of calcium in the intestine is estrogen-dependent, it can have different effects in postmenopausal women [29]. Additionally, previous studies have revealed that coffee or tea consumption might have a positive effect on BMD through other components that have estrogenic, antioxidant, and anti-inflammatory effects [30,31]. A recent study reported that prolonged tea intake was positively associated with BMD, but if the amount of tea consumed was above 5 cups/day (1 cup = 300 mL), the positive effect disappeared accordingly [32]. This is thought to be due to the increased amounts of caffeine that might offset the benefits of polyphenols and isoflavonoids in tea. Further studies are necessary to investigate the optimal amount of tea consumption related to bone health.

The strength of this study is that we used nationwide representative data from the KNHANES, and therefore, it could be generalized to some extent. To the best of our knowledge, this is the first investigation of green tea consumption and bone health among Korean women after menopause. Although BMD is highly affected by genetics, Korean data were not included in the existing meta-analysis. In addition, unlike previous studies that did not investigate the type of tea, we specified the impact of green tea on BMD and osteoporosis. Compared to other teas, the protective effects of green tea extracts on bone have been actively studied [33,34]. We have also added evidence that green tea may have benefits to preserve bone health.

However, this study had several limitations. First, the questionnaire was limited in terms of the duration of consumption. Only data on green tea and frequency of intake were available, and frequency was only investigated for up to three cups per day. In addition, because the duration of tea intake was not available, the accumulated quantitative effects could not be reflected accordingly. It was also difficult to determine the optimal amount of tea intake, as the intake of more than three cups per day was not measured in this study. Second, the calcium content of the water used to brew tea can influence the EGCG content of the tea, which promotes bone mineralization. Franks et al. reported that tea brewed with bottled or deionized water contained higher EGCG compared to tea brewed with tap water [35]. However, in this study, the data of the type of water used to brew tea was not available. Third, despite our consideration of coffee intake, it was difficult to accurately reflect the dietary intake of caffeine. Although there were many caffeine-containing beverages in the market, the questionnaire did not include information on other caffeine-containing beverages. Fourth, although we excluded the participants who had comorbidities that cause secondary osteoporosis, medical history of other causes of bone loss such as malabsorption due to gastric or intestinal inflammatory diseases, hyperparathyroidism, Cushing syndrome, and taking medications (e.g., glucocorticoids, anticonvulsants, and anticoagulants) was not available, and we could not adjust these confounding variables. Finally, our study was a cross-sectional analysis, and it was difficult to clearly understand the causal relationship.

## 5. Conclusions

In conclusion, we found an inverse relationship between green tea consumption and osteopenia and osteoporosis among postmenopausal Korean women. These results indicate that green tea intake might be beneficial for bone health. Further, prospective studies or clinical trials considering the duration, amount, and type of tea consumption are necessary to elucidate the effects of green tea on BMD and the risk of osteoporosis and fracture.

## Figures and Tables

**Table 1 nutrients-14-00087-t001:** General characteristics of study subjects based on their green tea intake.

Variable	Cups of Green Tea/Day (*n* = 3530)			*p*-Value
	None (*n* = 1893)	<1/Day (*n* = 1336)	1–3/Day (*n* = 301)	
Age (years)	65.47 ± 0.31	61.54 ± 0.29	60.93 ± 0.60	<0.001
BMI (kg/m^2^)	24.31 ± 0.09	24.14 ± 0.10	24.72 ± 0.25	0.071
Smoking, *n* (%)				0.007
Ex-smoker	90 (5.2)	32 (2.5)	13 (5.2)	
Current smoker	88 (5.3)	44 (3.0)	11 (4.1)	
Binge alcohol drinking, *n* (%)	47 (2.5)	53 (5.1)	18 (8.3)	<0.001
Regular exercise, *n* (%)	881 (48.1)	650 (52.5)	169 (59.1)	0.004
Educational level, *n* (%)				<0.001
≤Elementary school	1459 (74.8)	749 (53.2)	144 (52.4)	
Middle	205 (12.0)	223 (18.7)	69 (20.9)	
High	175 (11.3)	253 (18.7)	69 (20.5)	
≥College	39 (1.9)	110 (9.4)	18 (6.2)	
Household income, *n* (%)				<0.001
Low	531 (26.3)	277 (20.4)	67 (25.4)	
Lower middle	511 (26.5)	320 (25.0)	65 (20.2)	
Upper middle	446 (26.5)	361 (26.6)	72 (21.1)	
High	381 (20.7)	359 (28.0)	95 (33.3)	
25-hydroxy vitamin D (ng/mL)	18.32 ± 0.29	18.08 ± 0.29	18.40 ± 0.48	0.799
Calcium intake (mg/day)	393.58 ± 14.29	470.50 ± 10.53	502.73 ± 24.05	<0.001
Coffee intake (cup/day)	0.967 ± 0.03	0.932 ± 0.03	1.31 ± 0.07	<0.001
Total energy intake (kcal/day)	1487.93 ± 22.56	1669.42 ± 25.51	1618.89 ± 52.78	<0.001
Protein intake (g/day)	47.69 ± 0.86	57.81 ± 1.08	55.76 ± 2.26	<0.001
Age of menopause (years)	48.65 ± 0.15	49.42 ± 0.16	49.85 ± 0.30	<0.001
Duration after menopause	16.57 ± 0.38	11.963 ± 0.34	10.964 ± 0.72	<0.001
Hormone replacement therapy, *n* (%)	202 (10.9)	270 (20.8)	66 (18.6)	<0.001
Comorbidities, *n* (%)				<0.001
none	642 (33.2)	324 (26.4)	91 (34.3)	
1–2	758 (47.4)	556 (46.6)	119 (45.3)	
≥3	493 (19.3)	456 (27.0)	91 (20.4)	

Values are presented as means ± standard error or unweighted numbers (weighted %). Comorbidities: Diabetes mellitus, Chronic kidney disease, Rheumatoid arthritis, Thyroid disease, Cancer, and Liver cirrhosis.

**Table 2 nutrients-14-00087-t002:** Prevalence of osteoporosis and osteopenia with green tea intake.

	Cups of Green Tea/Day			*p*-Value
	None	<1/Day	1–3/Day	
Lumbar spine (*n* = 3232)				<0.001
Normal	326 (20.2)	318 (27.2)	97 (40.4)	
Osteopenia	795 (47.6)	591 (48.5)	119 (40.7)	
Osteoporosis	605 (32.2)	321 (24.4)	60 (18.9)	
Femur neck (*n* = 3470)				<0.001
Normal	305 (18.6)	328 (26.3)	98 (36.0)	
Osteopenia	1043 (54.9)	770 (57.9)	157 (52.5)	
Osteoporosis	512 (26.5)	218 (15.8)	39 (11.5)	
Total femur (*n* = 3470)				<0.001
Normal	989 (56.0)	869 (67.9)	210 (74.0)	
Osteopenia	755 (37.7)	405 (28.6)	79 (24.5)	
Osteoporosis	116 (6.3)	42 (3.5)	5 (1.4)	
Lumbar spine or Femur neck or Total femur (*n* = 3530)				
Normal	206 (10.9)	218 (16.6)	74 (27.7)	<0.001
Osteopenia	864 (47.9)	706 (53.2)	151 (49.5)	
Osteoporosis	823 (41.2)	412 (30.2)	76 (22.8)	

Values are presented as unweighted numbers (weighted%).

**Table 3 nutrients-14-00087-t003:** The crude and adjusted means of bone mineral density with green tea intake.

Bone Mineral Density (g/cm^2^)	Cups of Green Tea/Day			
	None Mean (95% CI)	<1/Day Mean (95% CI)	1–3/Day Mean (95% CI)	*p* for Trend
Lumbar spine (*n* = 986)				
Crude	0.78 (0.78–0.79)	0.82 (0.81–0.83)	0.85 (0.83–0.86)	<0.001
Model 1	0.79 (0.79–0.80)	0.81 (0.80–0.82)	0.83 (0.81–0.85)	0.001
Model 2	0.84 (0.82–0.86)	0.85 (0.83–0.87)	0.85 (0.83–0.88)	0.266
Model 3	0.84 (0.82–0.86)	0.85 (0.83–0.87)	0.85 (0.83–0.88)	0.326
Femur neck (*n* = 769)				
Crude	0.60 (0.60–0.61)	0.64 (0.63–0.65)	0.66 (0.64–0.67)	<0.001
Model 1	0.62 (0.61–0.62)	0.63 (0.62–0.63)	0.64 (0.63–0.65)	0.002
Model 2	0.64 (0.62–0.65)	0.64 (0.63–0.66)	0.65 (0.63–0.66)	0.314
Model 3	0.64 (0.62–0.65)	0.64 (0.63–0.66)	0.65 (0.63–0.66)	0.362
Total femur (*n* = 163)				
Crude	0.75 (0.75–0.76)	0.78 (0.78–0.80)	0.81 (0.80–0.83)	<0.001
Model 1	0.77 (0.76–0.77)	0.77 (0.77–0.78)	0.79 (0.78–0.81)	0.004
Model 2	0.78 (0.76–0.80)	0.78 (0.77–0.80)	0.79 (0.77–0.81)	0.417
Model 3	0.78 (0.76–0.80)	0.78 (0.77–0.80)	0.79 (0.77–0.81)	0.411

CI: confidence interval. Model 1: age-adjusted; Model 2: age, BMI, socioeconomic status, behavioral, hormonal, and nutritional factor adjusted; and Model 3: model 3 is adjusted for comorbidity in addition to model 2.

**Table 4 nutrients-14-00087-t004:** The odds ratios for osteopenia according to green tea intake.

Osteopenia	Cups of Green Tea/Day		
	None OR (95% CI)	<1/Day OR (95% CI)	1–3/Day OR (95% CI)
Lumbar spine (*n* = 1505)			
Crude OR	2.33 (1.70–3.21)	1.76 (1.25–2.49)	1
Model 1	2.02 (1.46–2.81)	1.76 (1.24–2.50)	1
Model 2	1.53 (1.09–2.15)	1.56 (1.09–2.23)	1
Model 3	1.54 (1.10–2.16)	1.63 (1.14–2.33)	1
Femur neck (*n* = 1970)			
Crude OR			1
Model 1	1.67 (1.20–2.32)	1.54 (1.11–2.15)	1
Model 2	1.46 (1.04–2.07)	1.40 (0.99–1.98)	1
Model 3	1.50 (1.06–2.13)	1.43 (1.01–2.04)	1
Total femur (*n* = 1239)			
Crude OR	2.03 (1.44–2.86)	1.26 (0.89–1.79)	1
Model 1	1.38 (0.95–2.00)	1.26 (0.78–1.81)	1
Model 2	1.08 (0.72–1.64)	1.04 (0.70–1.55)	1
Model 3	1.09 (0.72–1.65)	1.02 (0.68–1.52)	1
Lumbar spine or Femur neck or Total femur (*n* = 2144)			
Crude OR	2.44 (1.66–3.59)	1.79 (1.21–2.64)	1
Model 1	2.11 (1.46–3.07)	1.83 (1.25–2.67)	1
Model 2	1.81 (1.21–2.71)	1.77 (1.19–2.63)	1
Model 3	1.81 (1.20–2.71)	1.85 (1.23–2.77)	1

OR; odds ratio, CI; confidence interval. Model 1: age-adjusted; Model 2: age, BMI, socioeconomic status, behavioral, hormonal, and nutritional factor adjusted; and Model 3: model 3 is adjusted for comorbidity in addition to model 2.

**Table 5 nutrients-14-00087-t005:** The odds ratios for osteoporosis according to green tea intake.

Osteoporosis	Cups of Green Tea/Day		
	None OR (95% CI)	<1/Day OR (95% CI)	1–3/Day OR (95% CI)
Lumbar spine (*n* = 986)			
Crude OR	3.41 (2.19–5.29)	1.91 (1.24–2.95)	1
Model 1	2.27 (1.41–3.65)	1.92 (1.21–3.03)	1
Model 2	1.43 (0.87–2.36)	1.30 (0.80–2.13)	1
Model 3	1.44 (0.87–2.37)	1.37 (0.84–2.23)	1
Femur neck (*n* = 769)			
Crude OR	4.47 (2.76–7.22)	1.88 (1.15–3.06)	1
Model 1	2.51 (1.41–4.46)	1.99 (1.12–3.53)	1
Model 2	1.92 (0.99–3.71)	1.59 (0.83–3.05)	1
Model 3	1.91 (0.99–3.69)	1.63 (0.85–3.12)	1
Total femur (*n* = 163)			
Crude OR	5.77 (2.21–15.04)	2.64 (0.98–7.06)	1
Model 1	2.69 (0.93–7.74)	2.80 (0.93–8.34)	1
Model 2	2.24 (0.56–8.83)	2.79 (0.66–11.70)	1
Model 3	2.32 (0.56–9.51)	2.76 (0.63–12.08)	1
Lumbar spine or Femur neck or Total femur (*n* = 1311)			
Crude OR	4.56 (2.92–7.11)	2.21 (1.42–3.41)	1
Model 1	2.83 (1.76–4.56)	2.32(1.46–3.67)	1
Model 2	1.93 (1.14–3.26)	1.71 (1.02–2.86)	1
Model 3	1.91 (1.13–3.23)	1.82 (1.09–3.05)	1

OR; odds ratio, CI; confidence interval. Model 1: age-adjusted; Model 2: age, BMI, socioeconomic status, behavioral, hormonal, and nutritional factor adjusted; and Model 3: model 3 is adjusted for comorbidity in addition to model 2.

## Data Availability

The KNHANES data can be downloaded from the website https://knhanes.kdca.go.kr/knhanes/main.do (accessed on 7 December 2021).

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
