# Peer review of "Relationship between Regular Green Tea Intake and Osteoporosis in Korean Postmenopausal Women: A Nationwide Study"

_nutrients, 2021, doi:10.3390/nu14010087_

Round 1

Reviewer 1 Report

Thank you, this is a really good study, well described. You do explain the limitations but I think your conclusion is a touch too optimistic. The language isn’t quite perfect but that does not prevent one understanding at all. Overall though you have done a lot of good work here and I enjoyed reading it and understood  the rationale and methods. 

Author Response

Thank you, this is a really good study, well described. You do explain the limitations but I think your conclusion is a touch too optimistic. The language isn’t quite perfect but that does not prevent one understanding at all. Overall though you have done a lot of good work here and I enjoyed reading it and understood the rationale and methods. 

[Response] We agree with reviewer’s comments, and revised the sentence to be more objective in the discussion part (line 208-209 and line 292).

Line 208-209; green tea intake is beneficial for bone health among postmenopausal Korean women.

--> green tea intake is a positive association with bone health among postmenopausal Korean women.

Line 292; green tea intake has benefits on bone health.

--> green tea intake might be beneficial for bone health.

Reviewer 2 Report

The manuscript entitled "Relationship Between Regular Green Tea Intake and Osteoporosis in Korean Postmenopausal Women: A Nationwide Study" can be considered for publication in Nutrients after minor corrections:

• please make sure that all tables are placed on one page: now, some tables have headlines on one page and rest of the table is on another page;

• line 143-144: Phrase unclear: did You mean that inthe group that consumed more tea You found more women with normal BMD? Please rephrase this sentence;

• line 151: there is no "more consumption" should be rather consumed more or had highier consumtion of tea; rephrase this sentence;

• In disussion section author write about coffe only in case of coffeine content, but it is known tha coffe also contains ather active substances (also polyphenols) – please discuss lalso that point

• Authors write in results section about socioeconomical status and age of participants: how that influences concious choice of drinking coffe or tea or both drinks? - please discuss tis issue;

Author Response

please make sure that all tables are placed on one page: now, some tables have headlines on one page and rest of the table is on another page;

[Response] We apologize for our mistake, and we revised all tables to be placed on one page.

line 143-144: Phrase unclear: did You mean that in the group that consumed more tea You found more women with normal BMD? Please rephrase this sentence;

[Response] According to the reviewer’s comments, and we revised the sentence (line 148-149).

line 151: there is no "more consumption" should be rather consumed more or had highier consumtion of tea; rephrase this sentence;

[Response] According to the reviewer’s comments, and we revised the sentence (line 169).

In disussion section author write about coffe only in case of coffeine content, but it is known tha coffe also contains ather active substances (also polyphenols) – please discuss lalso that point

[Response] Thank you for this crucial comment. We added the sentence and relevant reference, “coffee contains polyphenols such as caffeic acid and chlorogenic acid, which are known to inhibit osteoclastogenesis and bone resorption”. (line 249-251, reference 27)

Authors write in results section about socioeconomical status and age of participants: how that influences concious choice of drinking coffe or tea or both drinks? - please discuss tis issue;

[Response] Thank you for this crucial comment. We performed further analysis to compare age, household income, and educational level according to coffee consumption (Table A).

Table A. Age and socioeconomic status based on coffee intake.

Variable

Cups of coffee/day (n=3,485)

P-value

None
(n=638)

<1/day
(n=913)

1-3/day
(n=1,934)

Age (years)

66.72±0.498

64.33±0.416

62.34±0.258

<0.001

Educational level, n (%)

<0.001

≤ Elementary school

488 (75.6)

619 (65.7)

1245 (61.1)

Middle

62 (11.4)

128 (15.3)

307 (16.4)

High

61 (8.9)

118 (13.1)

318 (17.4)

≥ College

28 (4.1)

53 (5.8)

86 (5.1)

Household income, n (%)

0.763

Low

173 (26.1)

228 (22.6)

474 (24.1)

Lower middle

163 (24.6)

227 (25.0)

506 (25.7)

Upper middle

161 (27.1)

230 (26.8)

488 (25.4)

High

141 (22.2)

228 (25.6)

466 (24.8)

Participants who reported drinking 1-3 cups of coffee a day were younger and had a higher education than those who did not. However, household income was not different among coffee groups. In this study, participants who were younger and had higher education and income levels consumed more green tea. This is presumably because people who have knowledge of the various health benefits of green tea or coffee consume more green tea or coffee than those who do not. According to previous consumer studies, health benefits are the main reason that consumers buy green tea or coffee (reference: Canonical Correlations between Benefit Sought and Selection Attributes of Green Tea Consumers, Korean J. Community Living Science, 2011, 2, 327-339 and The Effect of Coffee Consumption Motivation on the Future Coffee Consumption Intentions, Asia-Pacific Journal of Business Venturing and Entrepreneurship, 2013, 8, 129-144.)

We added the sentences related to green tea, and relevant reference in the discussion section (line 240-247, reference 26).

Reviewer 3 Report

The study presented by Lee et al. investigates the relationship between green tea consumption and bone mineral density among post-menopausal Korean women. The authors claim that tea intake is beneficial for their bone health. The study has large numbers on its side and the manuscript is well written.

I have some suggestions for improving the manuscript:

Page 2, lines 84-87. The causes of secondary osteoporosis are more numerous, as they include malabsorption due to gastric or intestinal inflammatory diseases, e.g. pancreatic insufficiency, or adverse effects of medications.

Page 3, lines 117-119. The authors only mentioned logistic regression models among the methods for comparing bone parameters across groups. However, when observing Tables 1 and 2 it seems that some comparison procedure between means has been used, such as analysis of variance; if so, this must be specified in paragraph 2.5 of the statistical analysis.

Page 4, Table 1. In some columns the sum of percentages is not equal to 100. Were there missing cases, and were the percentages affected by some weighing procedure? Moreover, for some continuous variables, three decimals seem a bit too much.

Page 6, lines 174-180. I guess the authors did not address the problem of how tea was brewed, and in particular the calcium content of the water, as discussed in a recent article by Franks et al. (Nutrients. 2019; 11(1): 80). Is it possible that the overall effect of tea consumption on bone metabolism was influenced also by the hardness of the water to prepare the infusion? What do the authors think?

English language is generally excellent; the only exceptions being represented by:
Page 1, line 24: “…disorder that the structure…”
Page 2, line 61: “comprised” instead of “consisted”

Author Response

POINT-BY-POINT RESPONSES TO REVIEWERS’ COMMENTS

Page 2, lines 84-87. The causes of secondary osteoporosis are more numerous, as they include malabsorption due to gastric or intestinal inflammatory diseases, e.g. pancreatic insufficiency, or adverse effects of medications.

[Response] We agree, and this is one of the limitations of our study. As we already mentioned, we excluded the participants who had comorbidities that cause secondary osteoporosis (diabetes mellitus, chronic kidney disease, rheumatoid arthritis, thyroid disease, cancer, and liver cirrhosis) (lines 84-86). However, medical history of other causes of bone loss such as malabsorption due to gastric or intestinal inflammatory diseases, hyperparathyroidism, Cushing syndrome, and taking medications (e.g., glucocorticoids, anticonvulsants, and anticoagulants) was not available in this study, therefore, we could not adjust these confounding variables. We added this issue in the discussion section (lines 292-297).

Page 3, lines 117-119. The authors only mentioned logistic regression models among the methods for comparing bone parameters across groups. However, when observing Tables 1 and 2 it seems that some comparison procedure between means has been used, such as analysis of variance; if so, this must be specified in paragraph 2.5 of the statistical analysis.

[Response] Thank you for this crucial comment. According to the reviewer’s comments, and we added the statistical methods for Table 1-3 (lines 114-116 and lines 119-121). General characteristics of study subjects were compared using one-way analysis of variance for continuous variables and chi-square test for categorical variables (Table 1 and 2). Crude and adjusted least-squares mean of BMD were estimated for each green tea group, and a multiple linear regression model was used for a trend test (Table 3).

Page 4, Table 1. In some columns the sum of percentages is not equal to 100. Were there missing cases, and were the percentages affected by some weighing procedure? Moreover, for some continuous variables, three decimals seem a bit too much.

[Response] The sum of percentages in the “comorbidities” variable is not equal to 100, because that was affected by complex sample analysis, as you expected. We performed statistical analysis again to check for errors, and the results were the same as those presented in our manuscript.

According to the reviewer’s comments, we revised three decimals to two decimals of continuous variables (Table 1).

Page 6, lines 174-180. I guess the authors did not address the problem of how tea was brewed, and in particular the calcium content of the water, as discussed in a recent article by Franks et al. (Nutrients. 2019; 11(1): 80). Is it possible that the overall effect of tea consumption on bone metabolism was influenced also by the hardness of the water to prepare the infusion? What do the authors think?

[Response] Thank you for this crucial comment. We agree, and we think that the calcium content of the water used to brew tea can influence the epigallocatechin gallate (EGCG) content of the tea, which is known to increase alkaline phosphatase activity and promotes bone mineralization. Franks et al. reported that tea brewed with bottled or deionized water contained higher EGCG compared to tea brewed with tap water. We added this issue in the discussion section (lines 285-289, reference 35).

English language is generally excellent; the only exceptions being represented by:
Page 1, line 24: “…disorder that the structure…”
Page 2, line 61: “comprised” instead of “consisted”

[Response] We apologize for our mistake, and we revised the sentences.

Line 24: “…disorder that the structure…”

--> disorder characterized by the destroyed structure of bone tissue and reduced bone density

Line 61: “comprised”

--> consisted
